# Multipole engineering by displacement resonance: a new degree of freedom of Mie resonance

Yu-Lung Tang [1,9], Te-Hsin Yen [1,9], Kentaro Nishida [1,9], Chien-Hsuan Li[1], Yu-Chieh Chen[1], Tianyue Zhang [2], Chi-Kang Pai [1], Kuo-Ping Chen[3,4], Xiangping Li [2] ✉, Junichi Takahara [5,6] ✉ & Shi-Wei Chu [1,7,8] ✉

The canonical studies on Mie scattering unravel strong electric/magnetic optical responses in nanostructures, laying foundation for emerging meta-photonic applications. Conventionally, the morphology-sensitive resonances hinge on the normalized frequency, i.e. particle size over wavelength, but non-paraxial incidence symmetry is overlooked. Here, through confocal reflection microscopy with a tight focus scanning over silicon nanostructures, the scattering point spread functions unveil distinctive spatial patterns featuring that linear scattering efficiency is maximal when the focus is misaligned. The underlying physical mechanism is the excitation of higher-order multipolar modes, not accessible by plane wave irradiation, via displacement resonance, which showcases a significant reduction of nonlinear response threshold, sign flip in all-optical switching, and spatial resolution enhancement. Our result fundamentally extends the century-old light scattering theory, and suggests new dimensions to tailor Mie resonances.

Over a hundred years ago, Gustav Mie gave the closed-form analytical solution of scattered fields from spherical geometries under plane wave excitations, thereby opening up configuring optically-induced resonances using nano-scatters[1]. Since then Mie scattering has established a kernel to study nanoscale light-matter interactions and drives the development of nanophotonics with far-reaching applications[2]. Unlike their plasmonic counterparts, high-index dielectric nanostructures with inherent low losses support an abundance of electric and magnetic multipole responses beyond the electric type[3]. Configuring the interference between multipolar resonances leads to appealing new phenomena such as directional scattering[4], photonics qubits[5], and nanolaser[6].

Conventionally, manipulations of Mie resonances have been achieved through modifying the nanostructure shape/size[7,8], tuning refractive indices via nonlinear effects[9,10] and tailoring the incident field profile with engineering the polarization and numerical aperture (NA) of incident light[11–13]. However, because canonical Mie resonances in spherical particles illuminated by plane waves had been considered dependent on only the dielectric permittivity and the normalized frequency (i.e. particle size over wavelength) for a long time, the position symmetry of the incidence light was largely overlooked until the recent observation of distinct transverse scattering by position-dependent excitation within a tightly focused cylindrical vector beam (CVB)[12,14,15]. This notion impulses

[1]Department of Physics, National Taiwan University, 1, Sec 4, Roosevelt Rd., Taipei 10617, Taiwan. [2]Guangdong Provincial Key Laboratory of Optical Fiber Sensing and Communications, Institute of Photonics Technology, Jinan University, Guangzhou 510632, China. [3]Institute of Imaging and Biomedical Photonics, National Yang Ming Chiao Tung University, 301 Gaofa 3rd Road, Tainan 711, Taiwan. [4]Institute of Photonics Technologies, National Tsing Hua University, 301 Gaofa 3rd Road, Hsinchu, Taiwan. [5]Graduate School of Engineering, Osaka University, 2-1 Yamadaoka, Suita, Osaka 565-0871, Japan. [6]Photonics Center, Graduate School of Engineering, Osaka University, 2-1 Yamadaoka, Suita, Osaka 565-0871, Japan. [7]Molecular Imaging Center, National Taiwan University, 1, Sec 4, Roosevelt Rd., Taipei 10617, Taiwan. [8]Brain Research Center, National Tsing Hua University, 101, Sec 2, Guangfu Road, Hsinchu 30013, Taiwan. [9]These authors contributed equally: Yu-Lung Tang, Te-Hsin Yen, Kentaro Nishida. ✉e-mail: xiangpingli@jnu.edu.cn; takahara@ap.eng.osaka-u.ac.jp; swchu@phys.ntu.edu.tw

our investigation to unveil unaddressed more general resonance phenomena hinging on the nanostructures and relative displacement excitation with respect to the focal spot, extending the previously reported conditions which all relied on the multipolar resonances by CVB with a polarization singularity for the spherical particles.

In our research, we discovered that a simple Gaussian beam is capable of exciting multiple Mie resonances, over the original excitation modes included in the Gaussian beam, by applying displacement resonance to the non-spherical particles. Through the platform of laser scanning microscopy, we experimentally and theoretically performed a comprehensive investigation on the displacement resonance of non-spherical silicon nanoparticles with Gaussian beam, covering size/position dependences of Mie resonances, laser scanning image formation, as well as influences on the photothermal nonlinear scattering. Our results offer insight into the physics of dielectric resonant nanostructure, proposing the method of efficient control of Mie resonances for various applications such as all-optical switching and nanomaterial imaging.

## Results

### Concept of displacement resonance

Conventional Mie theory analyzes the relationship of scattering intensity over relative frequency, i.e. particle width ($w$) over excitation wavelength ($\lambda$), under plane wave incidence. In the cases of $w/\lambda \ll 1$ and $w/\lambda \gg 1$, Rayleigh scattering and geometric optics govern, respectively. When $w/\lambda \sim 1$, multiple Mie resonance peaks appear. Here we aim to explore new dimensions of Mie resonance with focused excitation, including the normalized particle width ($w$) versus the focus spot size ($FWHM$), and displacement ($d$) of focus relative to $FWHM$. In analogy to Mie resonance, the idea of displacement resonance is introduced in Fig. 1, where the relative spot size $w/FWHM$ and relative displacement $d/FWHM$ are considered with laser scanning images. Figure 1a presents nine schematics of varying relative spot size $w/FWHM$ and relative displacement $d/FWHM$. In the vertical axis, we consider three cases. The $w/FWHM \ll 1$ case corresponds to the scenario of plane wave (PW) incidence or of a loosely focused laser beam, where the nano-scatterer senses negligible field gradient and PW Mie resonance is expected. The $w/FWHM \sim 1$ case means that the focal spot size is comparable to the

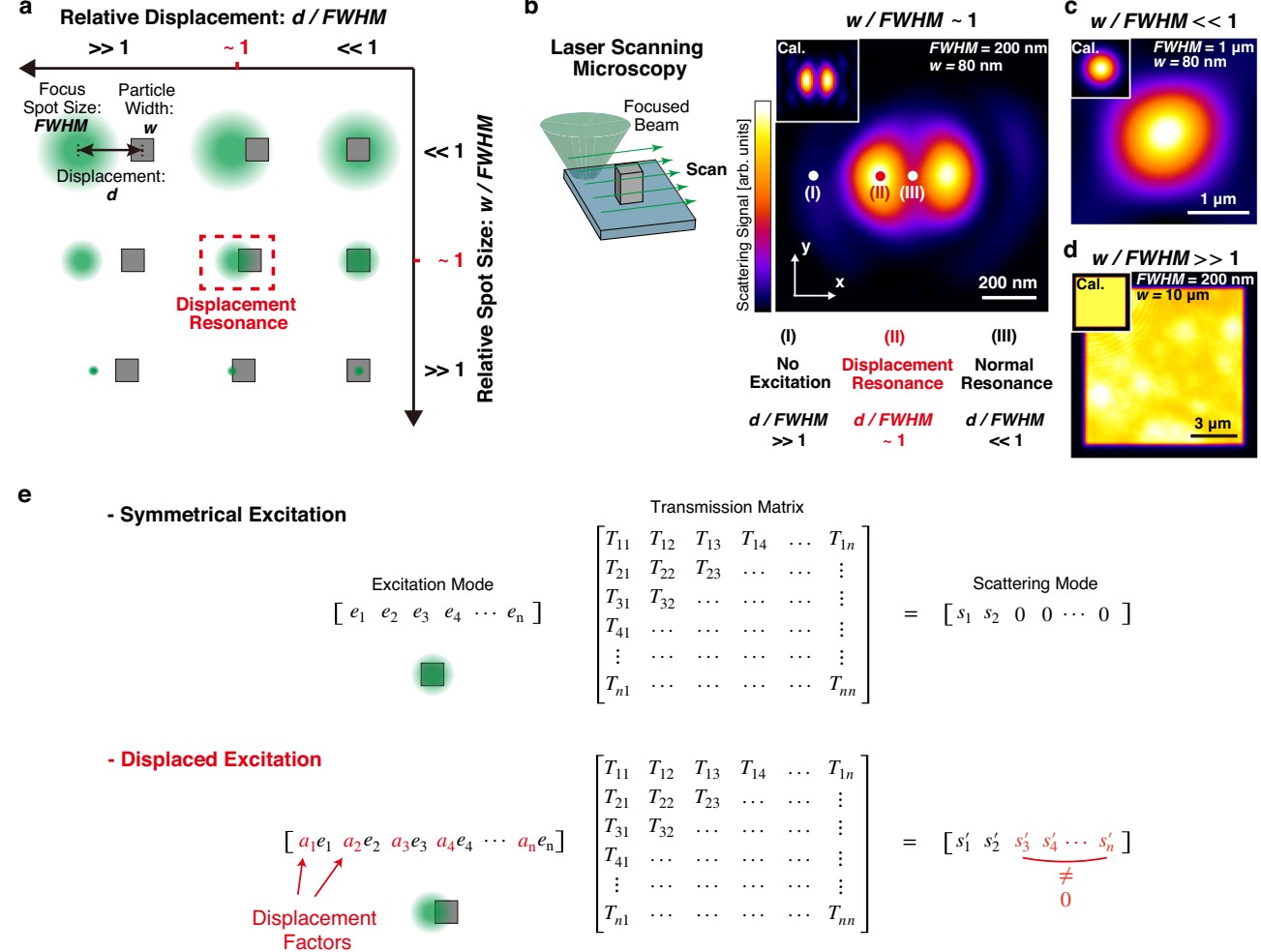

**Fig. 1 | Concept of displacement resonance. a** Schematic illustration of displacement resonance. The green circles and gray rectangular respectively indicate a focus spot with the size of *FWHM* and a dielectric particle with the width of *w*. The displacement *d* indicates the distance between the focus spot and the particle. We defined the condition of displacement resonance, in the case that both *d/FWHM* (relative displacement) and *w/FWHM* (relative spot size) are close to unity. **b** Example of displacement resonance. When a crystalline silicon particle diameter (*w* = 80 nm) is comparable to the focus spot size (*FWHM* = 200 nm), the particle shows maximum scattering intensity as the beam center is slightly displaced, forming a non-gaussian laser scanning image. Each pixel in the LSM image corresponds to a unique displacement between particle and beam center, as indicated by the (I), (II), and (III) spots in the scanning image. The upper left inset shows the simulated LSM image. **c** LSM image of the same silicon particle in **b**, but with a loosely focused beam (*FWHM* = 1 μm). **d** Laser scanning image of a thin silicon film with large diameter (*w* = 10 μm) under a tightly focused beam (*FWHM* = 200 nm). **e** Concept of transmission matrix model. The top and bottom figures respectively indicate the cases of symmetrical excitation (on-axis excitation, *d* = 0 nm) and displaced excitation (off-axis excitation, *d* ≠ 0).

particle size, typically under a tightly focused geometry when working with nanomaterials. The $w/FWHM \gg 1$ case represents thin film resonances, where the film size is much larger than the focal spot size.

In the horizontal axis, we also consider three cases. When the particle center aligns with the laser focus ($d/FWHM \ll 1$), the optical responses are similar to PW excited resonances. When the displacement is much larger than the spot size ($d/FWHM \gg 1$), there is no excitation with nanomaterials. Now the most interesting case occurs when both $d/FWHM$ and $w/FWHM$ are close to unity, where we termed it as displacement resonance.

Experimentally, we adopt confocal reflection laser scanning microscopy (LSM, see Supplementary Fig. 1 for details), which collects backscattering from nanostructures. The LSM provides a tightly focused spot with ~200 nm FWHM, as well as conveniently tunable displacement $d/FWHM$ via raster scan. The samples are monocrystalline silicon nanocuboids with 80 nm width (see "Methods" section) and a square silicon thin film with 10 μm width, both with 150 nm thickness.

In Fig. 1b, whose focal spot size is comparable to the particle size, i.e. $w/FWHM$ ~ 1, it depicts the scenario under a tightly focused geometry when working with nanomaterials. Apparently, the resulting image is far from a simple convolution between the laser spot and the nanoparticle. There are three displacement cases: (I) displacement is much larger than the spot size ($d/FWHM \gg 1$), so there is no excitation; (II) displacement is similar to the spot size ($d/FWHM$ ~ 1), creating the displacement resonance condition that unexpected bright spots appear; (III) the particle center aligns with the laser focus ($d/FWHM \ll 1$), so the optical responses should be similar to PW excited resonances, but intriguingly weaker than region (II).

Figure 1c is the same nanocuboid as in Fig. 1b, but now the objective NA reduces, resulting in a loosely focused spot size larger than 1 μm, thus approaching the $w/FWHM \ll 1$ condition. Surprisingly, the two images are dramatically different. The image of Fig. 1c is a solid Gaussian distribution that corresponds well to conventional imaging theory based on convolution[16]. This result is similar to dark-field microscopy observation under plane wave incidence (Supplementary Fig. 2 and Fig. 3). Nevertheless, the LSM image in Fig. 1b is not a solid circle, but splits into two horizontal lobes, featuring that the scattering maximum is no longer at the center, but occurs when the laser focus is NOT aligned with the nanostructure. This observation implies that the resonance condition varies at different excitation displacements.

Figure 1d is an LSM image of a thin film, whose width is much larger than the 200 nm FWHM of the focused laser spot, i.e. $w/FWHM \gg 1$. As we expected, the LSM image simply represents a thin film reflection pattern, as $d/FWHM$ varies from zero to much larger than unity. In order to obtain insights of the displacement resonance, we also developed an LSM simulation by solving Mie scattering fields sequentially across the nanostructure together with multipole decomposition analysis (MDA) to explain the unusual image patterns (see Supplementary Fig. 4 for details). Remarkably, the experimental LSM images of Fig. 1b–d are all reproduced well in the inset simulations.

The displacement resonance may be explained via the concept of the transmission matrix (T-matrix) model[15], as shown in Fig. 1e. In this model, the scattering cross-section of the nanostructure is represented as a T-matrix, containing partial scattering cross-sections representing various multipole modes. By expanding the excitation source with the same spherical vector wave basis as the T-matrix, the resulting scattering mode vector is expressed via matrix multiplication. When a Gaussian beam is aligned with the center, the excitation mode vector formulation is similar to plane wave incidence, as shown by the symmetrical excitation case in Fig. 1e. According to the translational addition theorem for Gaussian beam[17], when the beam is displaced, additional phase terms are added up to each of the excitation mode vector eigenmodes, thus modulating the output scattering vector. Considering the 80 nm particle in Fig. 1b as an example, for the

symmetrical excitation condition (point III of Fig. 1b), the scattering vector is mainly composed of the first two low-order elements $s_1$ and $s_2$, i.e. electric and magnetic dipole contributions. The bottom panel in Fig. 1e presents the case when the Gaussian beam is displaced, the additional phase terms (displacement factors in Fig. 1e) results in higher order components in the scattering vector. That is, additional resonant modes are excited under excitation beam displacement.

## Confirmation of displacement resonance by experiment and simulation

To present a systematic study of displacement resonance, we used an array of silicon nanocuboids whose height was 150 nm, and the lateral dimension ($w$) ranged from 80 to 280 nm in 10 nm steps (see Supplementary Fig. 5 for size verification, and Supplementary Movie 1 for animated pattern evolution). Figure 2a presents the simulated size- and displacement-dependent scattering cross-sections of the four leading multipoles (ED, MD, EQ, MQ) of the silicon nanocuboid array, under a tightly focused scheme ($FWHM = 200$ nm). The color here represents the percentage of each multipole's contribution to the total scattering cross-section. In the ED plot, most nanocuboids show maximal contribution at zero displacement, but the large particles ($w = 260–280$ nm) have maximum at 150 nm displacement. In the MD plot, both large and small particles exhibit maximum at zero displacement, and the mid-sized particles show interesting displacement resonance, extending to 200 nm displacement. Now in the case of the EQ plot, nearly all particles present displacement resonances. That is, no EQ is allowed for zero displacement with a focused laser beam (similar to plane wave incidence), but EQ emerges with focus displacement, in the range of 100–250 nm. Note that we also confirmed the generation of EQ mode due to the displacement resonance in the simulation result of the electric field distribution inside nanocuboid (see Supplementary Fig. 6). In the MQ plot, displacement resonance exists for the small particles, in particular the 80 nm one.

The heterogeneity of displacement resonance leads to a natural consequence that light-matter interaction during laser scanning no longer could be interpreted as a simple convolution between the nanostructure and the focal spot, as obviously shown in the size-dependent backward scattering LSM images of Fig. 2b (see Supplementary Fig. 7 for corresponding simulation). Here the laser is linearly polarized along the horizontal direction (see Supplementary Fig. 8 for laser polarization linearity check, and Supplementary Fig. 9 for the LSM images with different incident polarization direction). Comparing the first two images in Fig. 2b, i.e. the w = 80 nm and 90 nm ones, with merely 10 nm difference in width, the scanning image drastically changes from a dipole-like pattern to a solid circle, indicating the displacement resonance exhibits significant sensitivity on nanoscale dimensions. As width increases, the images gradually elongate in the $y$ direction, split into two lobes in the $w = 150$ nm one, return to solid circles with slight squash, and then become donut shapes in $w = 260–280$ nm. Apparently, it is not a rare case that the scattering maximum is not located at the center where the excitation focus aligns with the nanostructure. In addition, the longitudinal to vertical variation of these image profiles confirms that the depolarization effect of a Gaussian beam under tight focus does not play a significant role here. One more evidence is that a loosely focused Gaussian beam, in which depolarized effect is negligible, still generates displacement resonance effect when illuminating on a $w = 10$ μm film (see Supplementary Fig. 10 for spot size dependent scattering of the thin film and Supplementary Fig. 11 for the polarization dependency of the thin film).

To illustrate in more detail the underlying connection of these unusual LSM images in Fig. 2b to the displacement resonances in Fig. 2a, we present the MDA result along the line profiles of four representative silicon nanocuboids in Fig. 2c. For the $w = 80$ nm silicon nanocuboid, the two-peak image is the result of higher-order modes such as EQ/MQ emerging from the displacement resonance, along with

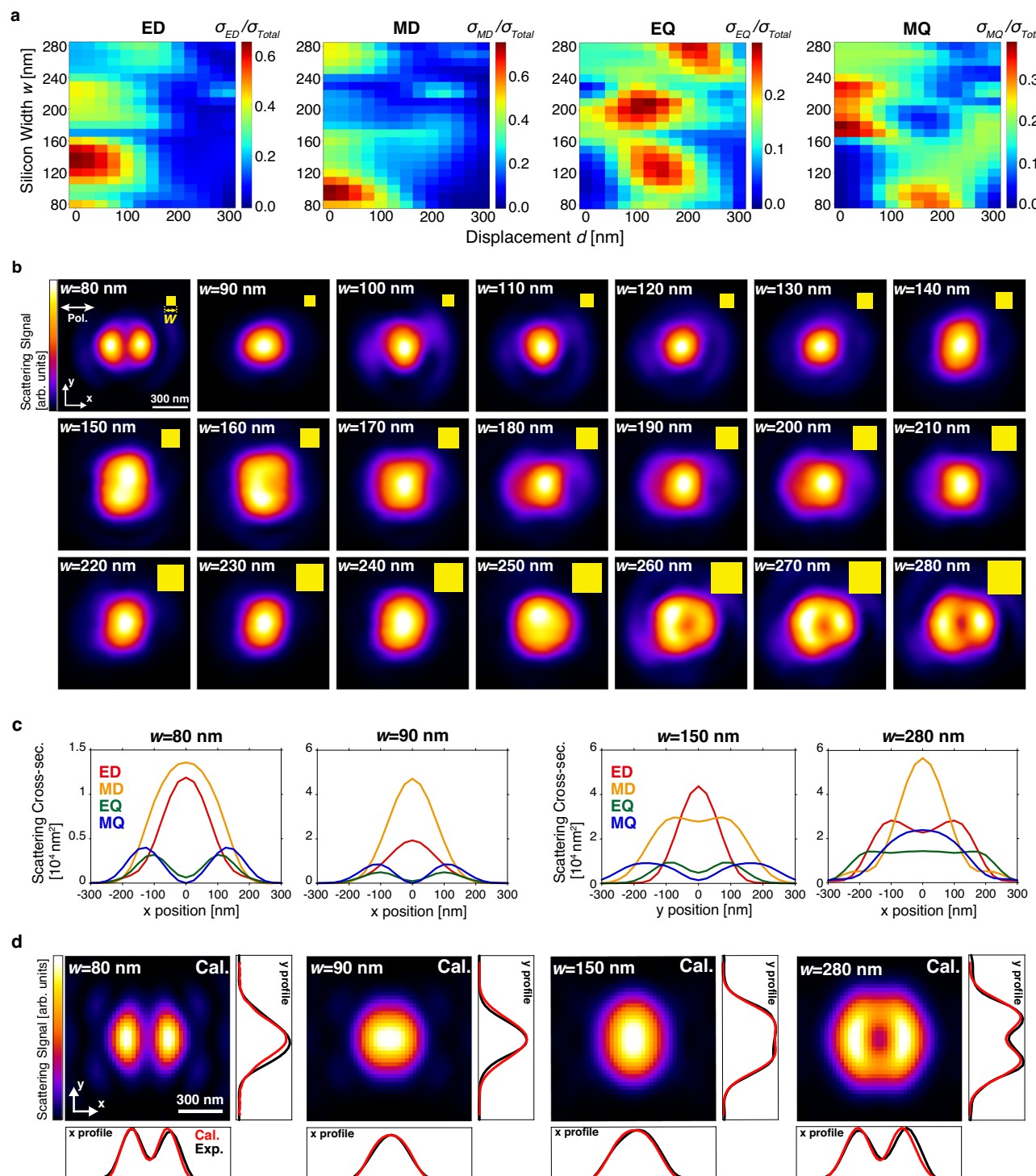

**Fig. 2 | Evidence of displacement resonance. a** Scattering cross-section of ED, MD, EQ and MQ modes calculated via MDA analysis. Each colormap shows the scattering cross-section contribution from the corresponding mode in specific size and displacement. The color represents the ratio of each multipole's contribution to the total scattering cross-section in percentage. **b** Size-dependent LSM images of silicon nanocuboids. The scattering intensities were normalized with individual images. The polarization direction of the electric field is horizontal. The yellow squares in the upper-right corner indicate the size of nanocuboids. Here the excitation intensity was enough low to avoid nonlinear scattering. **c** Multipole decomposition analysis (MDA) of four representative particles with focus displacements, to understand the image formation mechanism in **b**. **d** Simulated images and the corresponding line profiles of the four representative particles. The black lines are experimental data, red lines are calculated profiles, showing extraordinary agreement.

the destructive interference between ED and MD, which are out of phase to each other as shown in the Supplementary Fig. 12. On the other hand, for the $w$ = 90 nm one, MD is obviously stronger than ED, thus resulting in the solid circular shape in the image. As for the $w$ = 150 and 280 nm silicon nanocuboid, the dominating contributions to the elongated image in the $y$ direction and $x$ direction with the donut-shaped image come from the z-oriented MD and ED components at ~150 nm displacement, respectively (see also Supplementary Fig. 12).

The MDA of LSM demonstrates the capability of tailoring the multipolar resonances via displacement of a tightly focused beam in the spatial domain.

The simulated LSM images of four representative nanocuboids are shown in Fig. 2d, where the pronounced difference between $w = 80$ and 90 nm, the elongated image of $w = 150$ nm, and the donut-shaped image of $w = 280$ nm are all satisfactorily reproduced. The corresponding line profiles of the nanocuboids in the horizontal and vertical directions are shown at the bottom of Fig. 2d, manifesting quantitative agreement between experiment and theory.

### Enhancement of scattering nonlinearity via displacement resonance

Here we offer application examples of displacement resonance on enhancing optical nonlinearity, all-optical switching with sign flipping, and super-resolution imaging of silicon. Figure 3a presents the scattering images of the same array in Fig. 2b, but now with a higher excitation intensity to drive the nanoparticles into the nonlinear regime. We have recently reported Mie-resonance mediated giant photothermal nonlinear response of silicon nanoparticles[9,10], where laser-induced heating causes significant localized refractive index variation, and the subsequent thermo-optical effect in turn strongly modifies Mie resonance spectrum, resulting in 3–5 orders enhancement of photothermal nonlinearity compared to bulk silicon. Interestingly, for particles near 180 nm, Fig. 2b shows Gaussian-like images, but donut shapes are observed in Fig. 3a. That is, when the laser focus is aligned with the particle, the scattering intensity decreases with increasing excitation intensity, indicating sub-linear power dependence. On the contrary, for particles near 280 nm, Fig. 2b shows donut images, but in Fig. 3a the center signal strength becomes stronger, suggesting the existence of superlinear nonlinearity at the center. That is, from 180 nm to 280 nm, we witness the transition between sublinear to super-linear power dependence.

Figure 3b illustrates a surprising effect that the nonlinear power dependency enhances when the laser focus is NOT aligned with the nanostructure, exemplified with the 180 nm particle. The method of extracting nonlinear power dependency over the focal displacement from LSM images is explained in Supplementary Fig. 13 and Supplementary Movie 2. With on-axis excitation ($d = 0$ nm), the nanocuboid starts the sublinear scattering at the excitation intensity of 6.46 mW/$\mu$m$^2$, and sequentially shows the reverse saturation of scattering at 18.4 mW/$\mu$m$^2$. On the other hand, in the case that the nanocuboid is placed under the displaced excitation of 125 nm ($d = 125$ nm), the thresholds of excitation intensities for the sublinear and reverse saturated scattering becomes 5.65 mW/$\mu$m$^2$ and 12.6 mW/$\mu$m$^2$, respectively showing 1.14 times and 1.48 times reductions. This result indicates the enhancement of nonlinear response.

To expand the application scenario, Fig. 3c gives an example of all-optical switch sign flip with merely ~100 nm displacement of a focused laser beam. The left panel in Fig. 3c is the nonlinear power dependency sign flip of the 250 nm silicon nanocuboid, which presents superlinearity at zero displacement, gradually transiting to sub-linearity at 125 nm displacement. When a tightly focused beam is aligned with the 250 nm nanocuboid, i.e., $d = 0$, and the intensity is periodically tuned above and below the nonlinearity threshold, the right panel of Fig. 3c shows corresponding positive scattering deviations from the linear trend. Nevertheless, when the focused beam is shifted by 125 nm, with the same periodic power up and down, the scattering signals now exhibit negative deviation. The result demonstrates unprecedented capability of tuning all-optical switch sign via miniscale beam displacement.

Furthermore, Fig. 3d, e demonstrate that displacement resonance assists in resolution enhancement when combined with nonlinear scattering. Figure 3d is the displacement versus power dependency of the 280 nm particle, showing super-linearity at low displacement, but transforming into linear at 125 nm displacement. Figure 3e is the

corresponding images at low and high excitation intensities. Via subtraction, the linear parts between low and high panels are canceled, while the super-linearity results in a reduced spot in the center, manifesting the potential of resolution enhancement. Theoretically, the spatial resolution is unlimitedly improved by precisely tuning the excitation intensity and repeating the subtraction process[18]. However, practically, the achievable spatial resolution is limited by the degradation of SNR accompanied with the subtraction process. In this experiment, we were able to achieve 2–3 times improvement of spatial resolution compared with the conventional LSM system, by selecting the excitation intensity to keep enough SNR of the subtracted image.

## Discussion

In this work, we experimentally found unique LSM images of silicon nanostructures that cannot be explained by conventional imaging theory. We discovered that when focal spot size is comparable to the nanocuboid, backscattering maximum occurs when the focused excitation field is not aligned with the nanostructure, due to excitation of high-order multipoles that are not expected with plane-wave Mie theory. In analogy to Mie scattering that is based on relative frequency ($w/\lambda$), we theoretically verified the existence of new resonance modes due to relative spot size ($w/FWHM$) and relative displacement ($d/FWHM$). The displacement resonance may be understood in terms of transmission matrix[15], where the off-diagonal elements offer resonance modes that are not excited by plane wave incidence.

Our result fundamentally extends Mie theory with an additional degree of freedom in the displacement for multipolar resonance engineering, bringing a brand-new perspective to the nanophotonics community. For example, the displacement resonant scattering is very sensitive to the size of the nanostructure, and may serve to achieve high-precision structure identification through combining techniques such as spatial modulation spectroscopy[19]. With the additional degrees of freedom in displacement, novel scattering/heating behaviors[20] are expected, e.g. the surprising fact that nonlinear optical responses are more efficient and the sign of nonlinearity flips with displaced focus. The displacement-dependent photothermal nonlinearity provides additional contrast in the spatial domain, and via introducing super-resolution techniques, dramatic resolution enhancement is expected[21].

Furthermore, the concept of displacement resonance is not limited to Mie-resonant nanostructures. For example, an off-axis Gaussian beam is utilized to excite the whispering gallery mode resonance in the microspheres[22]. Furthermore, the material choice is not limited to silicon, and the wavelength is not limited to visible regime, but shall be general to various resonant conditions, including both metallic and dielectric materials across the electromagnetic spectrum. It is future work to explore the full potential of displaced excitation to efficiently control light-material coupling mode in order to extend the applications of resonant optical devices.

## Methods

### Experiment setup and sample information

We used a commercial confocal laser scanning system (FV-300, Olympus, Japan) combined with an inverted microscope (IX-71, Olympus, Japan) to obtain scanning images from silicon nanostructures. A schematic of our setup is shown in Supplementary Fig. 1. A 561 nm and 150 mW continuous-wave laser was adopted as our light source. The laser was sent through a home-built attenuator composed of a half-wave plate and a polarization beam splitter and then sent into the laser scanning system. Raster scanning was performed by a set of galvo mirrors at the speed of 3 $\mu$s/pixel, and the light was focused by either a high-NA objective (UPlanSApo ×100 oil, NA1.4, Olympus, Japan) or a low-NA one (UPlanSApo ×10, NA0.4, Olympus, Japan). The backward scattering signal was de-scanned by galvo mirrors and extracted with a 50/50 beam splitter, reflected by a mirror, passed through a confocal pinhole and collected by a built-in backward photomultiplier tube (PMT). To avoid

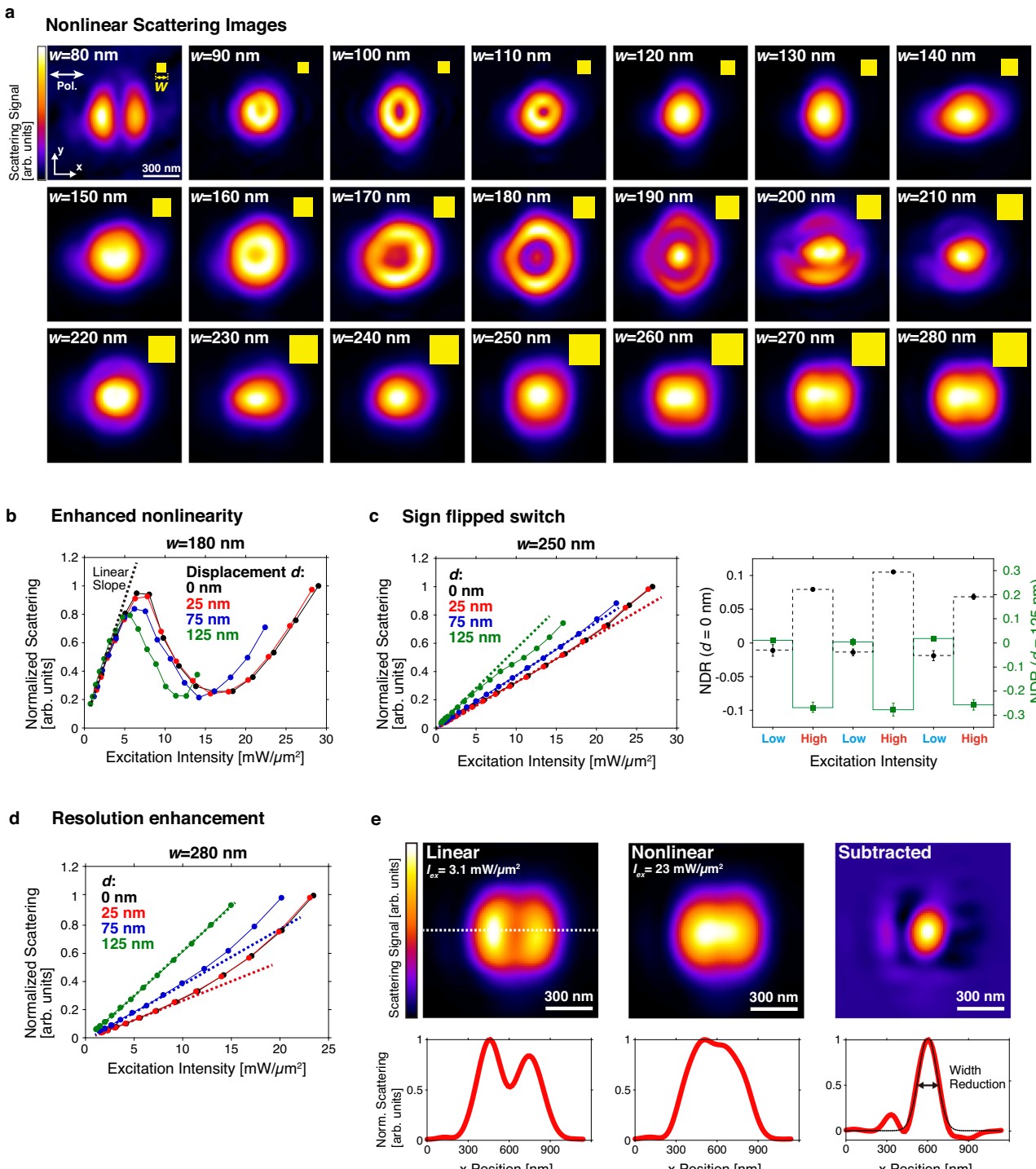

**Fig. 3 | Applications of displacement resonance on optical nonlinearity. a** Size-dependent LSM images of silicon nanocuboids that are driven into nonlinear regime under high laser intensity (24 mW/μm²). **b** Reduction of nonlinearity threshold via displacement resonance in the $w=180$ nanocuboid. Dashed line represents the linear extrapolation from the low excitation regime. **c** With the $w=250$ nm nanocuboid, the left panel presents that the sign of nonlinear response flips from super-linear to sub-linear as displacement increases, leading to a sign flip of all-optical switching in the right panel. Here the nonlinear deviation ratio (NDR) is defined by the deviation ratio of the measured scattering intensity over the extrapolated linear scattering intensity. Low and high excitation intensities in the right panel respectively are 2 and 22 mW/μm² for $d=0$ nm (black circular plot with dotted line) and $d=125$ nm (green rectangular plot with solid line) **d, e** Resolution enhancement with the $w=280$ nm nanocuboid. **d** is the displacement-dependent nonlinearity evolution. **e** shows linear, nonlinear, and subtracted LSM images, with corresponding line profiles, manifesting significant FWHM reduction after subtraction. The excitation intensities for linear and nonlinear images are respectively 3.1 and 23 mW/μm². The FWHM of the subtracted image profile is 225 nm, obtained by Gaussian fitting (black line).

the de-polarization effect of the beam-splitters and the scanning system, an additional half-wave plate is placed right under the objective to rotate the incident linear polarization.

Our sample is rectangular monocrystalline silicon nanocuboids with different widths ranging from 80 nm to 280 nm with a 10 nm step and a fixed height of 150 nm. The sample was sitting on a quartz substrate immersed in index matching oil. The 150-nm-thick monocrystalline silicon on a quartz substrate (Shin-Etsu Chemical Co., Ltd.) was fabricated by wafer bonding to silicon wafer surface. The nanostructure was made by electron beam lithography (ELS-7700T Elionix Inc.).

As for the experiment of silicon thin film, the setup remained the same for confocal backward scattering measurement. For forward scattering measurement, we used two different objectives (UPlanSApo ×10, NA0.4, and UPlanSApo ×40, NA0.95, Olympus, Japan) together with a dark-field condenser (U-DCD dry dark-field condenser, NA = 0.8–0.92, Olympus, Japan) and measured the forward scattering with photomultiplier tube in the transmission path.

### Simulation parameters

We used the RF module of FEM-based software COMSOL Multiphysics (COMSOL Inc.) to produce all of our simulation data. The schematic of the simulation setup is shown in Supplementary Fig. 4. The calculation domain was a sphere with a 1200 nm radius. The silicon nanocuboid was placed in the center of the calculation domain. The real and imaginary refractive indices of the silicon nanocuboid were set to 3.9786 and 0.02302, respectively, which were the experimentally measured values at the excitation wavelength of 561 nm by using an ellipsometer in our previous research[10]. The refractive index of the surrounding medium was set to $n = 1.518$ to correspond with immersion oil in the experiment. The calculation domain was surrounded by a perfect matching layer (PML) to eliminate boundary reflection. The thickness of PML was 600 nm. The collecting surface for the backward scattering signal was set at a distance of 200 nm in front of the PML. The area of the collecting surface reflects the NA of the objective lens (=1.40 in our research).

For illumination light, we assumed a linear polarized Gaussian electromagnetic wave focused by the objective lens with an NA of 1.40 to correspond with our experimental situation. The non-paraxial Gaussian electromagnetic wave was calculated based on the angular spectrum method[23] by using build-in function of COMSOL Multiphysics. The detailed mathematical formulation of the non-paraxial excitation beam is shown in the Supplementary Notes 1.

### Simulation of laser scanning image

We constructed the laser scanning simulation by sequentially solving the scattering problem (see Supplementary Notes 2 for detailed theoretical description), but with silicon nanocuboid located at different positions. For example, if the nanocuboid is located at the center of the calculation domain, the calculated backward scattering cross-section is identified as the central pixel in the scanning image; if the nanocuboid is located 100 nm away from the center in the x direction, the calculated backward scattering cross-section is identified as the pixel 100 nm distant from the center in the scanning image. Due to the symmetry of our rectangular sample and linear polarized electric field, only a quarter of the image is needed since the rest can be found by the mirror of the quarter. To prove the validity of this simulation model, we compared the simulated scanning profile with convolution calculation, since it is well known that the image is given by the convolution of object function and the amplitude point spread function (PSF) of the system for coherent signals[16].

## Data availability

The data of the main manuscript is provided in the Source Data file. Additional data related to this paper may be available upon request. Source data are provided with this paper.

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

## Acknowledgements

S.W.C. thanks Ministry of Science and Technology, Taiwan grant 112-2112-M-002-032-MY3, 112-2321-B-002-025, and 112-2119-M-002-022-

MBK; The Featured Areas Research Center Program (NTHU) and NTU Higher Education Sprout Project (NTU-112L8809) within the framework of the Higher Education Sprout Project co-funded by the MOST and the Ministry of Education, Taiwan (S.W.C.) for funding. J.T. thanks The Japan Society for the Promotion of Science (JSPS) KAKENHI Grant No. JP19H02630 and Core-to-Core Program "Advanced Nanophotonics in the Emerging Fields of Nanoimaging, Spectroscopy, Nonlinear Optics, Plasmonics/Meta-materials and Devices." X.L. thanks National Key R&D Program of China grant 2021YFB2802000 (XL). Sample fabrication was supported by the Nanotechnology Platform of MEXT, Grant Nos. JPMXP09F19OS0004 and JPMXP09F20OS0020. K.N. and J.T. acknowledge Shin-Etsu Chemical Co., Ltd. for providing high-quality SOQ wafers. S.W.C. and X.L. thank Dr. Yuri Kivshar for his insightful discussion.

## Author contributions

Y.L.T. and S.W.C. initiated this research project. Y.L.T., T.H.Y., C.H.L., and C.K.P. performed the experiments. J.T. prepared the sample for the experiment. Y.L.T., Y.C., T.Z., and X.L. worked on simulation investigation and theoretical discussion. Y.L.T., T.H.Y., and K.N. worked on the visualization of all results. S.W.C., J.T., and X.L. acquired the funding for this research project. Y.L.T., T.H.Y., and K.N. prepared an original draft of the manuscript. T.Z., X.L., K.P.C., J.T., and S.W.C. reviewed and edited the manuscript before the submission.

## Competing interests

The authors declare no competing interests.
