## [Peer Review File · Nature Communications]

Multipole engineering by displacement resonance: a new degree of freedom of Mie resonanceReviewer #1 (Remarks to the Author):

The manuscript "Non-paraxial displacement resonance: a new dimension to tailor Mie nanoresonators" by Yu-Lung Tang et al. investigates excitation of Mie modes in silicon nanostructures under the illumination by a displaced, tightly focused beam. The manuscript presents their experimental work using confocal reflection microscopy and support the results with numerical simulations. The work is technically correct and the idea well described. However, the idea of using spatially structured incident beams to selectively excite Mie resonances has been quite explored in the literature, including under displacement conditions. This configuration has been used, via analysis of the directional scattering, to measure nanometric displacements of the particles. The main novelty here seems to lie on the applications, namely its use to manipulate optical non-linearities. This, however, might not be sufficiently significant to grant publication in Nat. Comm. In this regards, I have some major concerns about the manuscript as listed below:

1. The authors cite ref. (11-14) wherein, the selective excitation or manipulation of different Mie multipoles has been investigated. In the present work, authors had explored the modification of Mie resonances under the effect of displacement of tightly focused beam w.r.t particle center. The major contribution or the novelty of this work is not very clear to me. Is it the non-linearity manipulation?
2. The authors have supported their experimental findings with simulation results. But they did not mention which method or software was being used. It seems FEM based software COMSOL was being used, but some more details allowing reproducibility are missing.
3. Authors mention that non-paraxial Gaussian beam was simulated using "angular spectrum method". Authors should specify whether the components in the longitudinal direction were being considered or not. These components cannot be ignored while studying field coupling to Mie nanostructures placed in the focal region of a tightly focused beam. A detailed description of simulation set up will help in better reproducibility.
4. In my opinion, a better insight into the physics is missing. Maybe the authors could consider a simple mathematical model (maybe using Mie theory under non-plane wave excitation?) and explain how symmetry breaking via displaced focused beam excites higher order Mie modes.
5. In the manuscript, authors had described application examples of displacement resonance on enhancing optical nonlinearity, all-optical switching with sign flipping, and super-resolution imaging of silicon. This seems to be the main novelty of the manuscript, but results are not correctly quantified; for example, what is the enhancement achieved in non-linear response? In case of super-resolution, what is the best achievable lateral resolution? Resolution was achieved by subtracting linear response from the non-linear response. The wall-plug efficiency of this approach seems too high.
6. The references in the supplementary are not properly cited. For example, "A tightly focused Gaussian electromagnetic wave, defined by the angular spectrum method (Ref 23), was irradiated..." The main text contains total 20 references only. It is ambiguous what "ref 23" refers to. The ideal practice is to provide the reference list of supplementary, locally there.

Overall, while interesting and technically correct, I have doubts on whether the presented work is significant and novel enough to be published in Nature Communications.

Reviewer #2 (Remarks to the Author):

The paper titled "Non-paraxial displacement resonance: a new dimension to tailor Mie nanoresonators" presents an interesting study on the morphology-sensitive resonances of Mie scattering. The authors use confocal reflection microscopy to demonstrate that non-paraxial incidence symmetry plays an important role in deterministic resonances, and the excitation of higher-order multipolar resonance modes emerges when the focus is displaced. This discovery

showcases a reduction of the nonlinear response threshold, sign flip in all-optical switching, and spatial resolution enhancement. The results are quite interesting and experimentally demonstrate the selectivity of the multipolar modes' excitation in simple geometry, showing an interesting interplay and mode competition between several Mie resonances with varied Q-factor. I believe that the paper is worth publishing in a high-profile journal like Nature Communications and brings an interesting vision of linear and nonlinear Mie scattering with resonant nanostructures. However, there are still some aspects the authors could improve upon:

- 1) They should provide more details on the multipole analysis used in the study and how it was performed.
- 2) It would be helpful to know whether the paraxial (Gaussian) approximation was used in the modeling and how the non-paraxial case would change the results.
- 3) The authors should provide more information on how the scattering cross-section was computed. Normally, the cross-section parameter is introduced for plane or infinite energy wave scattering, but the energy in a Gaussian beam is finite. Therefore, one can speak of scattering efficiency or simply the fraction of energy scattered.
- 4) The bottom panel of Fig. 2 (c) looks slightly redundant, and the authors do not discuss it properly in the text. Though the data shown there might be quite important for understanding the underlying physics, the authors do not discuss it properly in the text. For instance, the left panel shows only ED_x and MD_y components, while the right one shows z-components. It would be better if they could explain the message that should be taken from looking at it.
- 5) At this moment, it would also be helpful if the authors provided the mode content of the cubes, especially since the higher-order quadrupole modes appear to take an important role.
- 6) The NDR figure needs improvement as it is hardly readable and almost unresolved in black/white coloring.

Reply to the comments of Reviewer #1
and a summary of the changes made in the revised manuscript.

Changes in the revised manuscript as a response to the reviewers' comments are highlighted in red color and clarifications regarding the reviewer's comments are provided in blue color.

The manuscript “Non-paraxial displacement resonance: a new dimension to tailor Mie nanoresonators” by Yu-Lung Tang et al. investigates excitation of Mie modes in silicon nanostructures under the illumination by a displaced, tightly focused beam. The manuscript presents their experimental work using confocal reflection microscopy and support the results with numerical simulations. The work is technically correct and the idea well described. However, the idea of using spatially structured incident beams to selectively excite Mie resonances has been quite explored in the literature, including under displacement conditions. This configuration has been used, via analysis of the directional scattering, to measured nanometric displacements of the particles. The main novelty here seems to lie on the applications, namely its use to manipulate optical non-linearities. This, however, might not be sufficiently significant to grant publication in Nat. Comm. In this regards, I have some major concerns about the manuscript as listed below:

Response:

We appreciate the reviewer for carefully reviewing our manuscript. As the reviewer indicated, manipulation of optical nonlinearity is one of important applications in our research. Nevertheless, the main novelty of our work lies in the excitation of higher-order multipolar modes by using displacement excitation. Conventionally, canonical Mie resonances in spherical particles illuminated by plane waves were considered dependent on only the dielectric permittivity and the normalized frequency (i.e. particle size over wavelength) for a long time. Therefore, the position symmetry and illumination spot size of the incidence light were not regarded as an important parameter to control the Mie resonance modes. However, in our research, we discovered that higher-order Mie resonance modes are efficiently triggered in the case that both the displacement distance and focus spot size are close to the size of nanoparticles, which is in analogy to the condition of normalized frequency inducing Mie resonance. This idea will be impactful to the fundamental physics of light-matter interaction at nanoscale with dielectric materials.

Although structured beams (ref. 11-14) with a polarization singularity have been explored to excite higher-order Mie resonances, our work represents a fundamentally-different aspect by using a simple Gaussian beam with uniform polarizations but relative displacements to efficiently excite higher-order multipolar resonances, which completes the missing knowledge gap in well-known Mie resonances. Furthermore, we showcase that the undressed Mie resonance is revealed through laser scanning microscopy, which may offer additional technique platform to explore Mie-Optics dealing with nanoscale light-matter interactions. We have explicitly addressed the reviewer's concerns about the novelty of our work in the following point-by-point response, and also reinforced the description of the main manuscript to clarify the standpoint of our research with the past references.

Comments #1: The authors cite ref. (11-14) wherein, the selective excitation or manipulation of different Mie multipoles has been investigated. In the present work, authors had explored the modification of Mie resonances under the effect of displacement of tightly focused beam w.r.t particle center. The major contribution or the novelty of this work is not very clear to me. Is it the non-linearity manipulation?

Response:

We agree that the concept of manipulating Mie resonance modes with structured illumination has been reported in these Ref. [11-14], which we already included in the main manuscript. In Ref. 11 [Das, T., Phys. Rev. B 92, 241110, 2015] and Ref 12 [Manna, U. et al, J. Appl. Phys. 127, 033101, 2020], they proposed techniques to manipulate the Mie resonance modes by engineering the polarization and numerical aperture (NA) of incident light. In particular, in Ref. 13 [Woźniak, P., et al., Laser Photon. Rev. 9, 231–240, 2015] and Ref 14 [Neugebauer, M., et al., Nat. Commun. 7, 11286, 2016], they reported that specific Mie resonance modes are the sensitive relative position of cylindrical vector beams for nanoparticles, demonstrating the ability to achieve high precision position sensing. Therefore, we assume that Ref. 13 and Ref 14 are the main reasons that the reviewer argued “In the present work, authors had explored the modification of Mie resonances under the effect of displacement of tightly focused beam w.r.t particle center.”

Nevertheless, the physical mechanism of our technique is clearly distinguishable from previous works, which all relied on the excitation of multipolar resonances in spherical particles by cylindrical vector beams with spatially variant polarizations. The main novelty of our work is that a simple Gaussian beam with uniform polarizations is capable of exciting multiple Mie resonance modes, over the original excitable modes included in the Gaussian beam, by applying displacement resonance to nonspherical particles, which completes the missing knowledge and technique gap in current well-known Mie-Optics. In our research, by using laser scanning microscopy, which is a less regarded method in the nanophotonics field, we experimentally and theoretically performed comprehensive investigation on the displacement resonance of non-spherical nanoparticle with Gaussian beam, covering size/position dependences of Mie resonances, laser scanning image formation, as well as induction of photothermal nonlinear scattering. We believe that our research provides new important insight into the physics of dielectric nanophotonics, including the method of efficient control of Mie resonances for various applications such as optical switching and nanomaterial imaging.

Based on above discussion, we have reinforced our introduction in the 2nd paragraph of the page 3 in the manuscript as below:

“Conventionally, manipulations of Mie resonances have been achieved through modifying the nanostructure shape/size [Grahn, P., et al., Phys. Rev. B Condens. Matter 86, 035419, 2012; Zhang, X, et al., Nat. Mater. 7, 435–441, 2008], tuning refractive indices via nonlinear effects [Zhang, T. et al. Nat. Commun. 11, 1–9, 2020; Duh, Y.-S. et al., Nat. Commun. 11, 4101, 2020], and tailoring the incident field profile with engineering the polarization and numerical aperture (NA) of incident light [Das, T., Phys. Rev. B Condens. Matter 92, 241110, 2015; Manna, U. et al, J. Appl. Phys. 127, 033101, 2020; Woźniak, P., et al., Laser Photon. Rev. 9, 231–240, 2015]. However, because canonical Mie resonances in spherical particles illuminated by plane waves had been considered dependent on only the dielectric permittivity and the normalized frequency (i.e. particle size over wavelength) for a long time, the position symmetry of the incidence light was largely overlooked until the recent observation of distinct transverse scattering by position-dependent excitation within a tightly focused cylindrical vector beam (CVB) [Manna, U. et al, J. Appl. Phys. 127, 033101, 2020; Woźniak, P., et al., Laser Photon. Rev. 9, 231–240, 2015; Krasikov, S. et al., Phys. Rev. Applied 15, 024052, 2021]. This notion impules our investigation to unveil unaddressed more general resonance phenomena hinging on the nanostructures and relative displacement excitation with respect to the focal spot, extending the previously reported conditions which all relied on the multipolar resonances by CVB with a polarization singularity for the spherical particles.

In our research, we discovered that a simple Gaussian beam is capable of exciting multiple Mie resonances, over the original excitation modes included in the Gaussian beam, by applying displacement resonance to the nonspherical particles. By using the platform of laser scanning microscopy, we experimentally and theoretically performed comprehensive investigation on the displacement resonance of non-spherical silicon nanoparticle with Gaussian

beam, covering size/position dependences of Mie resonances, laser scanning image formation, as well as influences on the photothermal nonlinear scattering. Our research may provide a new important insight into the physics of dielectric resonant nanostructure, proposing the method of efficient control of Mie resonances for various applications such as optical switching and nanomaterial imaging.”

We also have modified the title of our manuscript to more specifically express the main point of our research, as “Multipole engineering by displacement resonance: a new degree of freedom of Mie resonance”.

Comments #2: The authors have supported their experimental findings with simulation results. But they did not mention which method or software was being used. It seems FEM based software COMSOL was being used, but some more details allowing reproducibility are missing.

Response:

Yes, the reviewer is correct that we use COMSOL Multiphysics, together with an RF module, to produce the simulation data. To improve the reproducibility of our work, we have added more detailed information on the simulation setup on COMSOL interface, as well as every physical constant used in our simulation, in the 1st paragraph of “Simulation parameters” section and the figure caption of Fig. S4 in the Supplementary Material, as below:

“We used the RF module of FEM-based software COMSOL Multiphysics (COMSOL Inc.) to produce all of our simulation data. The schematic of the simulation setup is shown in Fig. S4. The calculation domain was a sphere with a 1200 nm radius (r_{calc}). The silicon nanocuboid was placed in the center of the calculation domain. The real and imaginary refractive indices of the silicon nanocuboid were set to 3.9786 and 0.02302, respectively, which were the experimentally measured values at the excitation wavelength of 561 nm by using an ellipsometer in our previous research [Duh, et al., Nat. Comm. 11, 4101 (2020)]. The refractive index of the surrounding medium was set to $n = 1.518$ to correspond with immersion oil in the experiment. The calculation domain was surrounded by a perfect matching layer (PML) to eliminate boundary reflection. The thickness of PML (t_{PML}) was 600 nm. The collecting surface for the backward scattering signal was set at a distance of 200 nm in front of the PML. The area of the collecting surface reflects the NA of the objective lens (= 1.40 in our research).“

Figure S4: Cross-section view of the simulation environment. The central gray square indicates the silicon nanocuboid placed at the center of the calculation domain. The real and imaginary refractive indices of the silicon

nanocuboid were set to 3.9786 and 0.02302, respectively, which were the experimentally measured values at the excitation wavelength of 561 nm by using an ellipsometer in our previous research [Duh, et al., Nat. Comm. 11, 4101 (2020)]. The radius of the calculation domain (r_{calc}) is 1200 nm. The calculation domain is surrounded by a perfect matching layer (PML) with a thickness (t_{PML}) of 600 nm. The blue arc indicates the collecting surface, which reflects the NA of the objective lens ($=1.40$), for scattering signal collection in the backward direction. The medium has a refractive index of $n=1.518$ which corresponds to the refractive index of immersion oil.

Comments #3: Authors mention that non-paraxial Gaussian beam was simulated using “angular spectrum method”. Authors should specify whether the components in the longitudinal direction were being considered or not. These components cannot be ignored while studying field coupling to Mie nanostructures placed in the focal region of a tightly focused beam. A detailed description of simulation set up will help in better reproducibility.

Response:

We used a non-paraxial Gaussian beam in all of our calculations. Therefore, the components of the longitudinal components were considered. We have added the detailed description about the calculation procedure of non-paraxial Gaussian beam by using angular spectrum method as the “Calculation of non-paraxial excitation beam” in the page 3 of Supplementary Materials, as below:

“In our simulation, we used a non-paraxial Gaussian beam produced by the built-in COMSOL function of the angular spectrum method. In the angular spectrum method, the electric field $E(r)$ is calculated via the Helmholtz equation [G. P. Agrawal, et al., Phys. Rev. A. 27, pp. 1693–1695, 1983].

$$(\nabla^2 + k^2)\mathbf{E}(\mathbf{r}) = 0 \quad (\text{S1})$$

where the electric field $\mathbf{E}(\mathbf{r})$ separated into transverse E_t and longitudinal E_z components by using unit vectors $\mathbf{n}(\mathbf{r})$ and $\mathbf{z}(\mathbf{r})$,

$$\mathbf{E}(\mathbf{r}) = \mathbf{n}(\mathbf{r})E_t + \mathbf{z}(\mathbf{r})E_z \quad (\text{S2})$$

Here, complete solutions of E_t and E_z by expanding as a power series,

$$E_t = \psi e^{ikz} = \left(\sum_{n=0}^{\infty} f^{2n} \psi^{(2n)} \right) e^{ikz} \quad (\text{S3})$$

$$E_z = \phi e^{ikz} = \left(\sum_{n=0}^{\infty} f^{2n+1} \phi^{(2n+1)} \right) e^{ikz} \quad (S4)$$

where, f is perturbation parameter which defined as:

$$f = w_0/l \quad (S5)$$

where, w_0 is the beam width and l is the diffraction length. Each component of $\psi^{(n)}$ and $\phi^{(n)}$ in Eq.(S3) and Eq.(S4) are obtained by solving the following differential equations,

$$\left(2i \frac{\partial}{\partial \zeta} + \frac{\partial^2}{\partial \xi^2} + \frac{\partial^2}{\partial \eta^2} \right) \psi^{(0)} = 0 \quad (S6)$$

$$\left(2i \frac{\partial}{\partial \zeta} + \frac{\partial^2}{\partial \xi^2} + \frac{\partial^2}{\partial \eta^2} \right) \psi^{(2n)} = - \frac{\partial^2 \psi^{(2n-2)}}{\partial \zeta^2} \quad (S7)$$

$$\phi^{(0)} = i \frac{\partial \psi^{(0)}}{\partial \xi} \quad (S8)$$

$$\phi^{(2n+1)} = i \frac{\partial \psi^{(2n)}}{\partial \xi} + i \frac{\partial \phi^{(2n-1)}}{\partial \xi} \quad (S9)$$

where,

$$\zeta = z/l \quad (S10)$$

$$\xi = y/w_0 \quad (S11)$$

$$\eta = z/w_0 \quad (S12)$$

We applied the following boundary condition E_b at $z = 0$, which corresponds to the paraxial approximation of the Gaussian beam at the focal plane, to obtain the solution of Eq. S4 and S5,

$$\psi^{(2n)}(x, y, 0) = \begin{cases} E_b(x, y), & \text{if } n = 0 \\ 0, & \text{if } n \geq 1 \end{cases} \quad (S13)$$

$$E_b(x, y) = E_0 \exp\left[-\frac{x^2+y^2}{w_0^2} - ik \frac{x^2+y^2}{2R}\right] \quad (S14)$$

where R is the radius of curvature at $z = 0$. Then, the total electric field $\mathbf{E}(\mathbf{r})$ is obtained from Eq. S2, S4 and S5, by solving the equations numerically with the boundary condition, while taking into account the relative placement of sample and excitation within the calculation domain.”

Comments #4: In my opinion, a better insight into the physics is missing. Maybe the authors could consider a simple mathematical model (maybe using Mie theory under non-plane wave excitation?) and explain how symmetry breaking via displaced focused beam excites higher order Mie modes.

Response:

The displacement resonance is expressed by borrowing the concept of the transmission matrix (T-matrix) model [Krasikov, S. et al. Phys. Rev. Applied 15, 024052, 2021]. We have added the schematic to explain our idea as Fig. 1(e) of the main manuscript. In this model, the scattering cross-section of the nanostructure is represented as a T-matrix, containing partial scattering cross-sections representing various multipole modes. By expanding the excitation source with the same spherical vector wave basis as the T-matrix, the resulting scattering mode vector is expressed via matrix multiplication. When a Gaussian beam is aligned with the center, the excitation mode vector formulation is similar to plane wave incidence, as shown by the “symmetrical excitation” case in Fig. 1e. According to the translational addition theorem for Gaussian beam [Doicu, A. and Wriedt, *Appl. Opt.* **36**, 2971-2978, 1997], when the beam is displaced, additional phase terms are added up to each of the excitation mode vector eigenmodes, thus modulating the output scattering vector. Considering the 80 nm particle in Fig. 1b of the main manuscript as an example, for the “symmetrical excitation” condition (point III of Fig. 1b), the scattering vector is mainly composed of the first two low-order elements s_1 and s_2 , i.e. electric and magnetic dipole contributions. The bottom panel in Fig. 1e presents the case when the Gaussian beam is displaced, the additional phase terms (“displacement factors” in Fig. 1e) results in higher order components in the scattering vector. That is, additional resonant modes are excited under excitation beam displacement.

We have added the above discussion in the 3rd paragraph of page 6 in the main manuscript.

e

- Symmetrical Excitation

$$\begin{array}{c}
 \text{Excitation Mode} \\
 [e_1 \ e_2 \ e_3 \ e_4 \ \dots \ e_n] \\
 \text{[Green Square]}
 \end{array}
 \begin{array}{c}
 \text{Transmission Matrix} \\
 \begin{bmatrix}
 T_{11} & T_{12} & T_{13} & T_{14} & \dots & T_{1n} \\
 T_{21} & T_{22} & T_{23} & \dots & \dots & \vdots \\
 T_{31} & T_{32} & \dots & \dots & \dots & \vdots \\
 T_{41} & \dots & \dots & \dots & \dots & \vdots \\
 \vdots & \dots & \dots & \dots & \dots & \vdots \\
 T_{n1} & \dots & \dots & \dots & \dots & T_{nn}
 \end{bmatrix}
 \end{array}
 = \begin{array}{c}
 \text{Scattering Mode} \\
 [s_1 \ s_2 \ 0 \ 0 \ \dots \ 0]
 \end{array}$$

- Displaced Excitation

$$\begin{array}{c}
 [a_1 e_1 \ a_2 e_2 \ a_3 e_3 \ a_4 e_4 \ \dots \ a_n e_n] \\
 \uparrow \uparrow \\
 \text{Displacement Factors} \\
 \text{[Green Square]}
 \end{array}
 \begin{array}{c}
 \text{Transmission Matrix} \\
 \begin{bmatrix}
 T_{11} & T_{12} & T_{13} & T_{14} & \dots & T_{1n} \\
 T_{21} & T_{22} & T_{23} & \dots & \dots & \vdots \\
 T_{31} & T_{32} & \dots & \dots & \dots & \vdots \\
 T_{41} & \dots & \dots & \dots & \dots & \vdots \\
 \vdots & \dots & \dots & \dots & \dots & \vdots \\
 T_{n1} & \dots & \dots & \dots & \dots & T_{nn}
 \end{bmatrix}
 \end{array}
 = \begin{array}{c}
 [s'_1 \ s'_2 \ \underbrace{s'_3 \ s'_4 \ \dots \ s'_n}_{\neq 0}]
 \end{array}$$

Figure 1e: Concept of transmission matrix model. The top and bottom figures respectively indicate the cases of symmetrical excitation (on-axis excitation, $d=0$ nm) and displaced excitation (off-axis excitation, $d \neq 0$).

Comments #5: In the manuscript, authors had described application examples of displacement resonance on enhancing optical nonlinearity, all-optical switching with sign flipping, and super-resolution imaging of silicon. This seems to be the main novelty of the manuscript, but results are not correctly quantified; for example, what is the enhancement achieved in non-linear response? In case of super-resolution, what is the best achievable lateral resolution? Resolution was achieved by subtracting linear response from the non-linear response. The wall-plug efficiency of this approach seems too high.

Response:

Following the reviewer's comment, we have quantified each value as below:

a) Enhancement achieved in nonlinear response

We described the enhancement of nonlinear response by using the threshold of excitation intensity to start the nonlinear scattering. As shown in Fig 3b of the main manuscript, the overall graph shapes of scattering versus excitation remain the same; however, the required excitation intensity to induce nonlinear scattering decreases by increasing the displacement of the excitation beam. For example, with on-axis excitation ($d=0$ nm), the nanocuboid starts the sublinear scattering at the excitation intensity of $6.46 \text{ mW}/\mu\text{m}^2$, and sequentially show the reverse saturation of scattering at $18.4 \text{ mW}/\mu\text{m}^2$. On the other hand, in the case that the nanocuboid is placed under the displaced excitation of 125 nm ($d=125 \text{ nm}$), the thresholds of excitation intensities for the sublinear and reverse saturated scattering becomes $5.65 \text{ mW}/\mu\text{m}^2$ and $12.6 \text{ mW}/\mu\text{m}^2$, respectively showing 1.14 times and 1.48 times reductions. We have added the above discussion in the 2nd paragraph of page 12 in the main manuscript.

b) The best achievable lateral resolution

Theoretically, the spatial resolution is unlimitedly improved by precisely tuning the excitation intensity and repeating the subtraction process [Nawa, et al., APL Photonics 3, 080805 (2018)]. However, practically, the achievable spatial resolution is limited by the degradation of SNR accompanied with the subtraction process. In this experiment, we were able to achieve 2-3 times improvement of spatial resolution compared with the conventional LSM system, by selecting the excitation intensity to keep enough SNR of the subtracted image. We have added the above discussion in the 3rd paragraph of page 13 in the main manuscript.

c) Wall-plug efficiency

The reviewer's "wall-plug efficiency" may refer to how efficient we can induce photothermal nonlinear scattering, i.e. the threshold of excitation intensity. Overall in our paper, the threshold excitation intensity shares similar values around 10 mW/ μm^2 and certainly they are relatively low compared with the threshold intensities to induce other nonlinearity such as Kerr nonlinearity and SHG. This is because photothermal and thermo-optic effects of the silicon nanostructures are greatly enhanced by Mie resonances resulting in a few order improved effective nonlinear refractive index n_2 when compared to other nonlinear effects, as we reported in our previous work [Duh, et al., Nat. Comm. 11, 4101 (2020)].

Comments #6: The references in the supplementary are not properly cited. For example, "A tightly focused Gaussian electromagnetic wave, defined by the angular spectrum method (Ref 23), was irradiated..." The main text contains total 20 references only. It is ambiguous what "ref 23" refers to. The ideal practice is to provide the reference list of supplementary, locally there.

Response:

Following the reviewer's suggestion, we have provided a local reference list of Supplementary Materials.

Overall, while interesting and technically correct, I have doubts on whether the presented work is significant and novel enough to be published in Nature Communications.

Response:

We believe these revisions have improved the manuscript and should clarify the reviewers' concerns. We hope the manuscript is now suitable for publication in Nature Communications.

**Reply to the comments of Reviewer #2
and a summary of the changes made in the revised manuscript.**

The paper titled "Non-paraxial displacement resonance: a new dimension to tailor Mie nanoresonators" presents an interesting study on the morphology-sensitive resonances of Mie scattering. The authors use confocal reflection microscopy to demonstrate that non-paraxial incidence symmetry plays an important role in deterministic resonances, and the excitation of higher-order multipolar resonance modes emerges when the focus is displaced. This discovery showcases a reduction of the nonlinear response threshold, sign flip in all-optical switching, and spatial resolution enhancement. The results are quite interesting and experimentally demonstrate the selectivity of the multipolar modes' excitation in simple geometry, showing an interesting interplay and mode competition between several Mie resonances with varied Q-factor. I believe that the paper is worth publishing in a high-profile journal like Nature Communications and brings an interesting vision of linear and nonlinear Mie scattering with resonant nanostructures. However, there are still some aspects the authors could improve upon:

Response:

We appreciate the reviewer's positive comments on our work.

Comments #1: They should provide more details on the multipole analysis used in the study and how it was performed.

Response:

Following the reviewer's suggestion, we have added a detailed description about the method of multipole analysis in the 2nd paragraph of "Calculation of scattering cross-section and multipole analysis" section in page 5 of Supplementary Materials as below:

"In order to performed multipole decomposition analysis (MDA), we first calculated the electric field distribution of nanostructure in the FEM numerical simulation software COMSOL Multiphysics (COMSOL Inc.), and obtained the electric current distribution by using this equation:

$$\mathbf{J}_{\omega}(\mathbf{r}) = i\omega\epsilon_0(\epsilon_r - 1)\mathbf{E}_{\omega}(\mathbf{r}) \quad (\text{S15})$$

Then, by the approach of Ref [R. Alaei, *Opt. Commun.* 407, pp. 17-21, 2018], multipolar moments in Cartesian coordinate are obtained by integration of current density in the Fourier space. The exact expression of these integration are as follows:

$$p_\alpha = -\frac{1}{i\omega} \left\{ \int d^3\mathbf{r} J_\alpha^\omega j_0(kr) + \frac{k^2}{2} \int d^3\mathbf{r} \left[3(\mathbf{r} \cdot \mathbf{J}_\omega) r_\alpha - r^2 J_\alpha^\omega \right] \frac{j_2(kr)}{(kr)^2} \right\} \quad (\text{S16})$$

$$m_\alpha = \frac{3}{2} \int d^3\mathbf{r} (\mathbf{r} \times \mathbf{J}_\omega)_\alpha \frac{j_1(kr)}{kr} \quad (\text{S17})$$

$$Q_{\alpha\beta}^e = -\frac{3}{i\omega} \left\{ \int d^3\mathbf{r} \left[3(r_\beta J_\alpha^\omega + r_\alpha J_\beta^\omega) - 2(\mathbf{r} \cdot \mathbf{J}_\omega) \delta_{\alpha\beta} \right] \frac{j_1(kr)}{kr} \right. \\ \left. + 2k^2 \int \left[5r_\alpha r_\beta (\mathbf{r} \cdot \mathbf{J}_\omega) - (r_\alpha J_\beta + r_\beta J_\alpha) r^2 - r^2 (\mathbf{r} \cdot \mathbf{J}_\omega) \delta_{\alpha\beta} \right] \frac{j_3(kr)}{(kr)^3} \right\} \quad (\text{S18})$$

$$Q_{\alpha\beta}^m = 15 \int d^3\mathbf{r} \left\{ r_\alpha (\mathbf{r} \times \mathbf{J}_\omega)_\beta + r_\beta (\mathbf{r} \times \mathbf{J}_\omega)_\alpha \right\} \frac{j_2(kr)}{(kr)^2} \quad (\text{S19})$$

where j_l is the l th order spherical Bessel function of the first kind.”

Comments #2: It would be helpful to know whether the paraxial (Gaussian) approximation was used in the modeling and how the non-paraxial case would change the results.

Response:

In this research, we did not use paraxial approximation in the modeling as also discussed in our response for the 3rd comment of Reviewer #1. We have added a detailed description on the method to calculate the field distribution of the focus spot, in the section “Calculation of non-paraxial excitation beam” in the page 3 of Supplementary Materials.

Comments #3: The authors should provide more information on how the scattering cross-section was computed. Normally, the cross-section parameter is introduced for plane or infinite energy wave scattering, but the energy in a Gaussian beam is finite. Therefore, one can speak of scattering efficiency or simply the fraction of energy scattered.

Response:

Following the reviewer’s suggestion, we have added a description about the calculation procedure of scattering cross-sections in the 1st paragraph of “Calculations of scattering cross-section and multipole analysis” section in page 5 of Supplementary Materials as below:

“We calculated the scattering cross-section by first integrating the time averaged energy flow through the collection surface defined by an NA of objective lens. The integrated result is then divided by the incoming Poynting vector to obtain a scattering cross-section. [C. F. Bohren, et al., “Absorption and scattering of light by small particles”, Wiley Interscience, New York (1983)]. Note that this definition of scattering cross-section is equivalent

to the fraction of energy scattered. Therefore, it is also feasible to calculate the scattering efficiency by multiplying the geometric cross-sectional area projected onto the focal plane.”

Comments #4: The bottom panel of Fig. 2 (c) looks slightly redundant, and the authors do not discuss it properly in the text. Though the data shown there might be quite important for understanding the underlying physics, the authors do not discuss it properly in the text. For instance, the left panel shows only ED_x and MD_y components, while the right one shows z-components. It would be better if they could explain the message that should be taken from looking at it.

Response:

We agree with the reviewer that the bottom panel of Fig.2(c) might be redundant, and thus we moved these figures to Fig S12. In addition, we also added the detailed explanation of Fig. 2(c) to explain our message more clearly, in the 3rd paragraph of page 9 in the main manuscript as below:

“To illustrate in more detail the underlying connection of these unusual LSM images in Fig. 2b to the displacement resonances in Fig. 2a, we present the MDA result along the line profiles of four representative silicon nanocuboids in Fig. 2c. For the $w = 80$ nm silicon nanocuboid, the two-peak image is the result of higher-order modes such as EQ/MQ emerging from the displacement resonance, along with the destructive interference between ED and MD, which are out of phase to each other, as shown in the Fig S12 of Supplementary Materials. On the other hand, for the $w = 90$ nm one, MD is obviously stronger than ED, thus resulting in the solid circular shape in the image. As for the $w = 150$ and 280 nm silicon nanocuboid, the dominating contributions to the elongated image in the y direction and x direction with the donut-shaped image come from the z-oriented MD and ED components at ~ 150 nm displacement, respectively (See also Fig S12 of Supplementary Materials). The MDA of LSM demonstrates the capability of tailoring the multipolar resonances via displacement of a tightly focused beam in the spatial domain.”

Comments #5: At this moment, it would also be helpful if the authors provided the mode content of the cubes, especially since the higher-order quadrupole modes appear to take an important role.

Response:

We agree that it is helpful to provide the mode contents. In Fig. 2a, the ED/MD/EQ/MQ mode contributions versus displacement are presented. Following the reviewer’s suggestion, we added a visualization example of the field distribution inside the $w = 280$ nm silicon nanocuboid, as shown in the new Fig. S6 (copied below for reviewer’s convenience).

Figure S6: Calculations of the electric field distribution inside a $w=280$ nm silicon nanocuboid. (A) Calculation results on non-displacement excitation ($d = 0$ nm). The position of the focus spot center is $x = 0$ nm, $y = 0$ nm and $z = -75$ nm. The left figure is xz slice of electric field distribution, circulating current and electric field show magnetic dipole resonance. The right figure is yz slice of electric field distribution, two hotspots concentrated at center show electric dipole resonance. (B) Calculation results with displacement excitation ($d = 200$ nm). The position of the focus spot center is $x = -200$ nm, $y = 0$ nm and $z = -75$ nm. The xz slice show that original circulating current is disrupted due to asymmetry after beam displacement, which causes disappearance of magnetic dipole mode. The yz slice show that four hotspots at four corners appear, which correspond to electric quadrupole mode.

Comments #6: The NDR figure needs improvement as it is hardly readable and almost unresolved in black/white coloring.

Response:

To make more clearly distinguishable two graphs of NDR Figure (right panel of Fig 3(c)) in black/white coloring, we have changed the format of the $d=125$ nm graph, from dash lines to solid lines, and from circular plots to rectangular plots.

Reviewer #1 (Remarks to the Author):

I think the authors have done a good job in replying to all comments and concerns raised. Therefore, I believe the paper can now be accepted for publication.

Reviewer #2 (Remarks to the Author):

The authors fully addressed the main concerns which I have raised. Regarding the novelty of the results presented by the authors, to my knowledge there is no experimental demonstration of the displaced Gaussian beam excitation of Mie modes. This result may reconsider the experimental approaches to Mie modes spectroscopy and microscopy. I also agree with the author's interpretation of the origin of this effect. In this view, it might be valuable to refer to this theoretical paper (<https://onlinelibrary.wiley.com/doi/full/10.1002/lpor.202000528>) where authors explain the multipolar origin of transverse coupling of WGMs. Probably some theoretical results on the excitation of other high-Q Mie modes with Gaussian beam could add the impact to this paper, but my opinion is that the paper deserves publication in the Nature Communication journal.

Dear Editors,

Many thanks for your editorial efforts and referees' reports on our manuscript entitled "Multipole engineering by displacement resonance: a new degree of freedom of Mie resonance". We have made the required revision to address comments from the referees and the editorial office. Please find below our point-to-point responses to referees' comments and a summary of the changes made in the revised manuscript. We also provided a marked manuscript to show all the changes and corrections.

On behalf of the authors

Xiangping Li, Junichi Takahara & Shi-Wei Chu

Reply to the comments of Reviewer #1

Comments #1: I think the authors have done a good job in replying to all comments and concerns raised. Therefore, I believe the paper can now be accepted for publication.

Responses: We appreciate the reviewer's positive comments on our work.

Reply to the comments of Reviewer #2

Comments #1: The authors fully addressed the main concerns which I have raised. Regarding the novelty of the results presented by the authors, to my knowledge there is no experimental demonstration of the displaced Gaussian beam excitation of Mie modes. This result may reconsider the experimental approaches to Mie modes spectroscopy and microscopy. I also agree with the author's interpretation of the origin of this effect. In this view, it might be valuable to refer to this theoretical paper (<https://onlinelibrary.wiley.com/doi/full/10.1002/lpor.202000528>) where authors explain the multipolar origin of transverse coupling of WGMs. Probably some theoretical results on the excitation of other high-Q Mie modes with Gaussian beam could add the impact to this paper, but my opinion is that the paper deserves publication in the Nature Communication journal.

Responses: We appreciate the reviewer for acknowledging our revised manuscript. We have added the new reference suggested by the reviewer as Reference 22 in the 2nd paragraph of page 12 the main manuscript, as below:

(Main manuscript, page 12, 2nd paragraph) "Furthermore, the concept of displacement resonance is not limited to Mie-resonant nanophotonic structures. For example, an off-axis Gaussian beam is utilized to excite the whispering gallery mode resonance in the microspheres (Ref. 22, Xavier Zambrana-Puyalto, et al., "Excitation Mechanisms of Whispering Gallery Modes with Direct Light Scattering", Laser Photon. Rev., 15, 5, 2021). Furthermore, the material choice is not limited to silicon, and the wavelength range is not limited to visible, but shall be general to various resonant conditions, including both metallic and dielectric materials across the electromagnetic spectrum. It is future work to explore the full potential of

displaced excitation to efficiently control light-material coupling mode in order to extend the applications of resonant optical devices.”